# Non-Local Recurrent Network for Image Restoration

Ding Liu[1],  Bihan Wen[1],  Yuchen Fan[1],  Chen Change Loy[2],  Thomas S. Huang[1]

[1]University of Illinois at Urbana-Champaign    [2]Nanyang Technological University

{dingliu2, bwen3, yuchenf4, t-huang1}@illinois.edu   ccloy@ntu.edu.sg

## Abstract

Many classic methods have shown non-local self-similarity in natural images to be an effective prior for image restoration. However, it remains unclear and challenging to make use of this intrinsic property via deep networks. In this paper, we propose a non-local recurrent network (NLRN) as the first attempt to incorporate non-local operations into a recurrent neural network (RNN) for image restoration. The main contributions of this work are: (1) Unlike existing methods that measure self-similarity in an isolated manner, the proposed non-local module can be flexibly integrated into existing deep networks for end-to-end training to capture deep feature correlation between each location and its neighborhood. (2) We fully employ the RNN structure for its parameter efficiency and allow deep feature correlation to be propagated along adjacent recurrent states. This new design boosts robustness against inaccurate correlation estimation due to severely degraded images. (3) We show that it is essential to maintain a confined neighborhood for computing deep feature correlation given degraded images. This is in contrast to existing practice [41] that deploys the whole image. Extensive experiments on both image denoising and super-resolution tasks are conducted. Thanks to the recurrent non-local operations and correlation propagation, the proposed NLRN achieves superior results to state-of-the-art methods with many fewer parameters. The code is available at `https://github.com/Ding-Liu/NLRN`.

## 1   Introduction

Image restoration is an ill-posed inverse problem that aims at estimating the underlying image from its degraded measurements. Depending on the type of degradation, image restoration can be categorized into different sub-problems, *e.g.*, image denoising and image super-resolution (SR). The key to successful restoration typically relies on the design of an effective regularizer based on image priors. Both local and non-local image priors have been extensively exploited in the past. Considering image denoising as an example, local image properties such as Gaussian filtering and total variation based methods [31] are widely used in early studies. Later on, the notion of self-similarity in natural images draws more attention and it has been exploited by non-local-based methods, *e.g.*, non-local means [2], collaborative filtering [8], joint sparsity [27], and low-rank modeling [15]. These non-local methods are shown to be effective in capturing the correlation among non-local patches to improve the restoration quality.

While non-local self-similarity has been extensively studied in the literature, approaches for capturing this intrinsic property with deep networks are little explored. Recent convolutional neural networks (CNNs) for image restoration [10, 20, 28, 49] achieve impressive performance over conventional approaches but do not explicitly use self-similarity properties in images. To rectify this weakness, a few studies [23, 30] apply block matching to patches before feeding them into CNNs. Nevertheless, the block matching step is isolated and thus not jointly trained with image restoration networks.

In this paper, we present the first attempt to incorporate non-local operations in CNN for image restoration, and propose a non-local recurrent network (NLRN) as an efficient yet effective network

with non-local module. First, we design a non-local module to produce reliable feature correlation for self-similarity measurement given severely degraded images, which can be flexibly integrated into existing deep networks while embracing the benefit of end-to-end learning. For high parameter efficiency without compromising restoration quality, we deploy a recurrent neural network (RNN) framework similar to [21, 35, 36] such that operations with shared weights are applied recursively. Second, we carefully study the behavior of non-local operation in deep feature space and find that limiting the neighborhood of correlation computation improves its robustness to degraded images. The confined neighborhood helps concentrate the computation on relevant features in the spatial vicinity and disregard noisy features, which is in line with conventional image restoration approaches [8, 15]. In addition, we allow message passing of non-local operations between adjacent recurrent states of RNN. Such inter-state flow of feature correlation facilitates more robust correlation estimation. By combining the non-local operation with typical convolutions, our NLRN can effectively capture and employ both local and non-local image properties for image restoration.

It is noteworthy that recent work has adopted similar ideas on video classification [41]. However, our method significantly differs from it in the following aspects. For each location, we measure the feature correlation of each location only in its neighborhood, rather than throughout the whole image as in [41]. In our experiments, we show that deep features useful for computing non-local priors are more likely to reside in neighboring regions. A larger neighborhood (the whole image as one extreme) can lead to inaccurate correlation estimation over degraded measurements. In addition, our method fully exploits the advantage of RNN architecture - the correlation information is propagated among adjacent recurrent states to increase the robustness of correlation estimation to degradations of various degrees. Moreover, our non-local module is flexible to handle inputs of various sizes, while the module in [41] handles inputs of fixed sizes only.

We introduce NLRN by first relating our proposed model to other classic and existing non-local image restoration approaches in a unified framework. We thoroughly analyze the non-local module and recurrent architecture in our NLRN via extensive ablation studies. We provide a comprehensive comparison with recent competitors, in which our NLRN achieves state-of-the-art performance in image denoising and SR over several benchmark datasets, demonstrating the superiority of the non-local operation with recurrent architecture for image restoration.

## 2   Related Work

Image self-similarity as an important image characteristic has been used in a number of non-local-based image restoration approaches. The early works include bilateral filtering [38] and non-local means [2] for image denoising. Recent approaches exploit image self-similarity by imposing sparsity [27, 44]. Alternatively, similar image patches are modeled with low-rankness [15], or by collaborative Wiener filtering [8, 47]. Neighborhood embedding is a common approach for image SR [5, 37], in which each image patch is approximated by multiple similar patches in a manifold. Self-example based image SR approaches [14, 12] exploit the local self-similarity assumption, and extract LR-HR exemplar pairs merely from the low-resolution image across different scales to predict the high-resolution image. Similar ideas are adopted for image deblurring [9].

Deep neural networks have been prevalent for image restoration. The pioneering works include a multilayer perceptron for image denoising [3] and a three-layer CNN for image SR [10]. Deconvolution is adopted to save computation cost and accelerate inference speed [34, 11]. Very deep CNNs are designed to boost SR accuracy in [20, 22, 24]. Dense connections among various residual blocks are included in [39]. Similarly CNN based methods are developed for image denoising in [28, 49, 50, 26]. Block matching as a preprocessing step is cascaded with CNNs for image denoising [23, 30]. Besides CNNs, RNNs have also been applied for image restoration while enjoying the high parameter efficiency [21, 35, 36].

In addition to image restoration, feature correlations are widely exploited along with neural networks in many other areas, including graphical models [51, 4, 17], relational reasoning [32], machine translation [13, 40] and so on. We do not elaborate on them here due to the limitation of space.

## 3   Non-Local Operations for Image Restoration

In this section, we first present a unified framework of non-local operations used for image restoration methods, *e.g.*, collaborative filtering [8], non-local means [2], and low-rank modeling [15], and we discuss the relations between them. We then present the proposed non-local operation module.

## 3.1 A General Framework

In general, a non-local operation takes a multi-channel input $\boldsymbol{X} \in \mathbb{R}^{N \times m}$ as the image feature, and generates output feature $\boldsymbol{Z} \in \mathbb{R}^{N \times k}$. Here $N$ and $m$ denote the number of image pixels and data channels, respectively. We propose a general framework with the following formulation:

$$\boldsymbol{Z} = \mathrm{diag}\{\delta(\boldsymbol{X})\}^{-1}\, \Phi(\boldsymbol{X})\, \boldsymbol{G}(\boldsymbol{X})\,. \tag{1}$$

Here, $\Phi(\boldsymbol{X}) \in \mathbb{R}^{N \times N}$ is the non-local correlation matrix, and $\boldsymbol{G}(\boldsymbol{X}) \in \mathbb{R}^{N \times k}$ is the multi-channel non-local transform. Each row vector $\boldsymbol{X}_i$ denotes the local features in location $i$. $\Phi(\boldsymbol{X})_i^j$ represents the relationship between the $\boldsymbol{X}_i$ and $\boldsymbol{X}_j$, and each row vector $\boldsymbol{G}(\boldsymbol{X})_j$ is the embedding of $\boldsymbol{X}_j$.[1] The diagonal matrix $\mathrm{diag}\{\delta(\boldsymbol{X})\} \in \mathbb{R}^{N \times N}$ normalizes the output at each $i$-th pixel with normalization factor $\delta_i(\boldsymbol{X})$.

## 3.2 Classic Methods

The proposed framework works with various classic non-local methods for image restoration, including methods based on low-rankness [15], collaborative filtering [8], joint sparsity [27], as well as non-local mean filtering [2].

Block matching (BM) is a commonly used approach for exploiting non-local image structures in conventional methods [15, 8, 27]. A $q \times q$ spatial neighborhood is set to be centered at each location $i$, and $\boldsymbol{X}_i$ reduces to the image patch centered at $i$. BM selects the $K_i$ most similar patches ($K_i \ll q^2$) from this neighborhood, which are used jointly to restore $\boldsymbol{X}_i$. Under the proposed non-local framework, these methods can be represented as

$$\boldsymbol{Z}_i = \frac{1}{\delta_i(\boldsymbol{X})} \sum\nolimits_{j \in \mathbb{C}_i} \Phi(\boldsymbol{X})_i^j\, \boldsymbol{G}(\boldsymbol{X})_j\,, \quad \forall i\,. \tag{2}$$

Here $\delta_i(\boldsymbol{X}) = \sum_{j \in \mathbb{C}_i} \Phi(\boldsymbol{X})_i^j$ and $\mathbb{C}_i$ denotes the set of indices of the $K_i$ selected patches. Thus, each row $\Phi(\boldsymbol{X})_i$ has only $K_i$ non-zero entries. The embedding $\boldsymbol{G}(\boldsymbol{X})$ and the non-zero elements vary for non-local methods based on different models. For example, in WNNM [15], $\sum_{j \in \mathbb{C}_i} \Phi(\boldsymbol{X})_i^j\, \boldsymbol{G}(\boldsymbol{X})_j$ corresponds to the projection of $\boldsymbol{X}_i$ onto the group-specific subspace as a function of the selected patches. Specifically, the subspace for calculating $\boldsymbol{Z}_i$ is spanned by the eigenvectors $\boldsymbol{U}_i$ of $\boldsymbol{X}_{\mathbb{C}_i}^T \boldsymbol{X}_{\mathbb{C}_i}$. Thus $\boldsymbol{Z}_i = \boldsymbol{X}_{\mathbb{C}_i} \boldsymbol{U}_i \mathrm{diag}\{\sigma\} \boldsymbol{U}_i^T$, where $\mathrm{diag}\{\sigma\}$ is obtained by applying the shrinkage function associated with the weighted nuclear norm [15] to the eigenvalues of $\boldsymbol{X}_{\mathbb{C}_i}^T \boldsymbol{X}_{\mathbb{C}_i}$. We show the generalization about more classic non-local image restoration methods in the supplementary material.

Except for the *hard* block matching, other methods, *e.g.*, the non-local means algorithm [2], apply *soft* block matching by calculating the correlation between the reference patch and each patch in the neighborhood. Each element $\Phi(\boldsymbol{X})_i^j$ is determined only by each $\{\boldsymbol{X}_i, \boldsymbol{X}_j\}$ pair, so $\Phi(\boldsymbol{X})_i^j = \phi(\boldsymbol{X}_i, \boldsymbol{X}_j)$, where $\phi(\cdot)$ is determined by the distance metric. In [2], weighted Euclidean distance with Gaussian kernel is applied as the metric, such that $\phi(\boldsymbol{X}_i, \boldsymbol{X}_j) = \exp\{-\|\boldsymbol{X}_i - \boldsymbol{X}_j\|_{2,a}^2 / h^2\}$. Besides, identity mapping is directly used as the embedding in [2], *i.e.*, $\boldsymbol{G}(\boldsymbol{X})_j = \boldsymbol{X}_j$. In this case, the non-local framework in (1) reduces to

$$\boldsymbol{Z}_i = \frac{1}{\delta_i(\boldsymbol{X})} \sum\nolimits_{j \in \mathbb{S}_i} \exp\{-\frac{\|\boldsymbol{X}_i - \boldsymbol{X}_j\|_{2,a}^2}{h^2}\} \boldsymbol{X}_j\,, \quad \forall i, \tag{3}$$

where $\delta_i(\boldsymbol{X}) = \sum_{j \in \mathbb{S}_i} \exp\{-\|\boldsymbol{X}_i - \boldsymbol{X}_j\|_{2,a}^2 / h^2\}$ and $\mathbb{S}_i$ is the set of indices in the neighborhood of $\boldsymbol{X}_i$. Note that both $a$ and $h$ are constants, denoting the standard deviation of Gaussian kernel, and the degree of filtering, respectively [2]. It is noteworthy that the cardinality of $\mathbb{S}_i$ for soft BM is much larger than that of $\mathbb{C}_i$ for hard BM, which gives more flexibility of using feature correlations between neighboring locations.

The conventional non-local methods suffer from the drawback that parameters are either fixed [2], or obtained by suboptimal approaches [8, 27, 15], *e.g.*, the parameters of WNNM are learned based on the low-rankness assumption, which is suboptimal as the ultimate objective is to minimize the image reconstruction error.

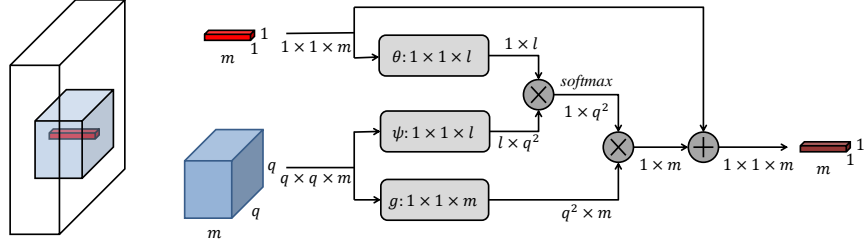

**Figure 1:** An illustration of our non-local module working on a single location. The white tensor denotes the deep feature representation of an entire image. The red fiber is the features of this location and the blue tensor denotes the features in its neighborhood. $\theta$, $\psi$ and $g$ are implemented by $1 \times 1$ convolution followed by reshaping operations.

### 3.3 The Proposed Non-Local Module

Based on the general non-local framework in (1), we propose another soft block matching approach and apply the Euclidean distance with linearly embedded Gaussian kernel [41] as the distance metric. The linear embeddings are defined as follows:

$$\Phi(\boldsymbol{X})_i^j = \phi(\boldsymbol{X}_i, \boldsymbol{X}_j) = \exp\{\theta(\boldsymbol{X}_i)\psi(\boldsymbol{X}_j)^T\}, \quad \forall i, j, \tag{4}$$

$$\theta(\boldsymbol{X}_i) = \boldsymbol{X}_i \boldsymbol{W}_\theta, \quad \psi(\boldsymbol{X}_i) = \boldsymbol{X}_i \boldsymbol{W}_\psi, \quad \boldsymbol{G}(\boldsymbol{X})_i = \boldsymbol{X}_i \boldsymbol{W}_g, \quad \forall i. \tag{5}$$

The embedding transforms $\boldsymbol{W}_\theta$, $\boldsymbol{W}_\phi$, and $\boldsymbol{W}_g$ are all learnable and have the shape of $m \times l$, $m \times l$, $m \times m$, respectively. Thus, the proposed non-local operation can be written as

$$\boldsymbol{Z}_i = \frac{1}{\delta_i(\boldsymbol{X})} \sum_{j \in \mathbb{S}_i} \exp\left\{\boldsymbol{X}_i \boldsymbol{W}_\theta \boldsymbol{W}_\psi^T \boldsymbol{X}_j^T\right\} \boldsymbol{X}_i \boldsymbol{W}_g, \quad \forall i, \tag{6}$$

where $\delta_i(\boldsymbol{X}) = \sum_{j \in \mathbb{S}_i} \phi(\boldsymbol{X}_i, \boldsymbol{X}_j)$. Similar to [2], to obtain $\boldsymbol{Z}_i$, we evaluate the correlation between $\boldsymbol{X}_i$ and each $\boldsymbol{X}_j$ in the neighborhood $\mathbb{S}_i$. More choices of $\phi(\boldsymbol{X}_i, \boldsymbol{X}_j)$ are discussed in Section 5.

The proposed non-local operation can be implemented by common differentiable operations, and thus can be jointly learned when incorporated into a neural network. We wrap it as a non-local module by adding a skip connection, as shown in Figure 1, since the skip connection enables us to insert a non-local module into any pre-trained model, while maintaining its initial behavior by initializing $\boldsymbol{W}_g$ as zero. Such a module introduces only a limited number of parameters since $\theta$, $\psi$ and $g$ are $1 \times 1$ convolutions and $m = 128, l = 64$ in practice. The output of this module on each location only depends on its $q \times q$ neighborhood, so this operation can work on inputs of various sizes.

**Relation to Other Methods:** Recent works have combined non-local BM and neural networks for image restoration [30, 23, 41]. Lefkimmiatis [23] proposed to first apply BM to noisy image patches. The hard BM results are used to group patch features, and a CNN conducts a trainable collaborative filtering over the matched patches. Qiao *et al.* [30] combined similar non-local BM with TNRD networks [7] for image denoising. However, as conventional methods [8, 27, 15], these works [23, 30] conduct hard BM directly over degraded input patches, which may be inaccurate over severely degraded images. In contrast, our proposed non-local operation as soft BM is applied on learned deep feature representations that are more robust to degradation. Furthermore, the matching results in [23] are isolated from the neural network, similar to the conventional approaches, whereas the proposed non-local module is trained jointly with the entire network in an end-to-end manner.

Wang *et al.* [41] used similar approaches to add non-local operations into neural networks for high-level vision tasks. However, unlike our approach, Wang *et al.* [41] calculated feature correlations throughout the whole image. which is equivalent to enlarging the neighborhood to the entire image in our approach. We empirically show that increasing the neighborhood size does not always improve image restoration performance, due to the inaccuracy of correlation estimation over degraded input images. Hence it is imperative to choose a neighborhood of a proper size to achieve best performance for image restoration. In addition, the non-local operation in [41] can only handle input images of fixed size, while our module in (6) is flexible to various image sizes. Finally, our non-local module, when incorporated into an RNN framework, allows the flow of correlation information between adjacent states to enhance robustness against inaccurate correlation estimation. This is a new unique formulation to deal with degraded images. More details are provided next.

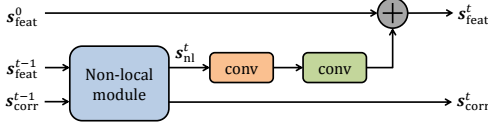

**Figure 2:** An illustration of the transition function $f_{\text{recurrent}}$ in the proposed NLRN.

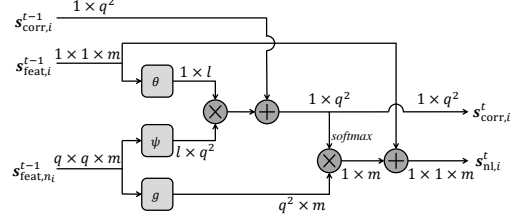

**Figure 3:** The operations for a single location $i$ in the non-local module used in NLRN.

# 4 Non-Local Recurrent Network

In this section, we describe the RNN architecture that incorporates the non-local module to form our NLRN. We adopt the common formulation of an RNN, which consists of a set of states, namely, input state, output state and recurrent state, as well as transition functions among the states. The input, output, and recurrent states are represented as $x$, $y$ and $s$ respectively. At each time step $t$, an RNN receives an input $x^t$, and the recurrent state and the output state of the RNN are updated recursively as follows:

$$s^t = f_{\text{input}}(x^t) + f_{\text{recurrent}}(s^{t-1}), \qquad y^t = f_{\text{output}}(s^t), \tag{7}$$

where $f_{\text{input}}$, $f_{\text{output}}$, and $f_{\text{recurrent}}$ are reused at every time step. In our NLRN, we set the following:

- $s^0$ is a function of the input image $I$.
- $x^t = 0$, $\forall t \in \{1, \ldots, T\}$, and $f_{\text{input}}(0) = 0$.
- The output state $y^t$ is calculated only at the time $T$ as the final output.

We add an identity path from the very first state which helps gradient backpropagation during training [35], and a residual path of the deep feature correlation between each location and its neighborhood from the previous state. Hence, $s^t = \{s^t_{\text{feat}}, s^t_{\text{corr}}\}$, and $s^t = f_{\text{recurrent}}(s^{t-1}, s^0)$, $\forall t \in \{1, \ldots, T\}$, where $s^t_{\text{feat}}$ denotes the feature map in time $t$ and $s^t_{\text{corr}}$ is the collection of deep feature correlation. For the transition function $f_{\text{recurrent}}$, a non-local module is first adopted and is followed by two convolutional layers, before the feature $s^0$ is added from the identity path. The weights in the non-local module are shared across recurrent states just as convolutional layers, so our NLRN still keeps high parameter efficiency as a whole. An illustration is displayed in Figure 2.

It is noteworthy that inside the non-local module, the feature correlation for location $i$ from the previous state, $s^{t-1}_{\text{corr},i}$, is added to the estimated feature correlation in the current state before the *softmax* normalization, which enables the propagation of correlation information between adjacent states for more robust correlation estimation. The details can be found in Figure 3. The initial state $s^0$ is set as the feature after a convolutional layer on the input image. $f_{\text{output}}$ is represented by another single convolutional layer. All layers have 128 filters with $3 \times 3$ kernel size except for the non-local module. Batch normalization and ReLU activation function are performed ahead of each convolutional layer following [18]. We adopt residual learning and the output of NLRN is the residual image $\hat{I} = f_{\text{output}}(s^T)$ when NLRN is unfolded $T$ times. During training, the objective is to minimize the mean square error $\mathcal{L}(\hat{I}, \tilde{I}) = \frac{1}{2}||\hat{I} + I - \tilde{I}||^2$, where $\tilde{I}$ denotes the ground truth image.

**Relation to Other RNN Methods:** Although RNNs have been adopted for image restoration before, our NLRN is the first to incorporate non-local operations into an RNN framework with correlation propagation. DRCN [21] recursively applies a single convolutional layer to the input feature map multiple times without the identity path from the first state. DRRN [35] applies both the identity path and the residual path in each state, but without non-local operations, and thus there is no correlation information flow across adjacent states. MemNet [36] builds dense connections among several types of memory blocks, and weights are shared in the same type of memory blocks but are different across various types. Compared with MemNet, our NLRN has an efficient yet effective RNN structure with shallower effective depth and fewer parameters, but obtains better restoration performance, which is shown in Section 5 in detail.

# 5 Experiments

**Dataset**: For image denoising, we adopt two different settings to fairly and comprehensively compare with recent deep learning based methods [28, 23, 49, 36]: (1) As in [7, 49, 23], we choose as the

training set the combination of 200 images from the *train* set and 200 images from the *test* set in the Berkeley Segmentation Dataset (BSD) [29], and test on two popular benchmarks: Set12 and Set68 with $\sigma = 15, 25, 50$ following [49]. (2) As in [28, 36], we use as the training set the combination of 200 images from the *train* set and 100 images from the *val* set in BSD, and test on Set14 and the BSD *test* set of 200 images with $\sigma = 30, 50, 70$ following [28, 36]. In addition, we evaluate our NLRN on the Urban100 dataset [19], which contains abundant structural patterns and textures, to further demonstrate the capability of using image self-similarity of our NLRN. The training set and test set are strictly disjoint and all the images are converted to gray-scale in each experiment setup. For image SR, we follow [20, 35, 36] and use a training set of 291 images where 91 images are proposed in [46] and other 200 are from the BSD *train* set. We adopt four benchmark sets: Set5 [1], Set14 [48], BSD100 [29] and Urban100 [19] for testing with three upscaling factors: $\times 2$, $\times 3$ and $\times 4$. The low-resolution images are synthesized by bicubic downsampling.

**Training Settings**: We randomly sample patches whose size equals the neighborhood of non-local operation from images during training. We use flipping, rotation and scaling for augmenting training data. For image denoising, we add independent and identically distributed Gaussian noise with zero mean to the original image as the noisy input during training. We train a different model for each noise level. For image SR, only the luminance channel of images is super-resolved, and the other two color channels are upscaled by bicubic interpolation, following [20, 21, 35]. Moreover, the training images for all three upscaling factors: $\times 2$, $\times 3$ and $\times 4$ are upscaled by bicubic interpolation into the desired spatial size and are combined into one training set. We use this set to train one single model for all these three upscaling factors as in [20, 35, 36].

We use Adam optimizer to minimize the loss function. We set the initial learning rate as 1e-3 and reduce it by half five times during training. We use Xavier initialization for the weights. We clip the gradient at the norm of $0.5$ to prevent the gradient explosion which is shown to empirically accelerate training convergence, and we adopt 16 as the minibatch size during training. Training a model takes about 3 days with a Titan Xp GPU. For non-local module, we use circular padding for the neighborhood outside input patches. For convolution, we pad the boundaries of feature maps with zeros to preserve the spatial size of feature maps.

## 5.1 Model Analysis

In this section, we analyze our model in the following aspects. First, we conduct the ablation study of using different distance metrics in the non-local module. Table 1 compares instantiations including Euclidean distance, dot product, embedded dot product, Gaussian, symmetric embedded Gaussian and embedded Gaussian when used in NLRN of 12 unfolded steps. Embedded Gaussian achieves the best performance and is adopted in the following experiments.

We compare the NLRN with its variants in terms of PSNR in Table 2. We have a few observations. First, the same model with untied weights performs worse than its weight-sharing counter-part. We speculate that the model with untied weights is prone to model over-fitting and suffers much slower training convergence, both of which undermine its performance. To investigate the function of non-local modules, we implement a baseline RNN with the same parameter number of NLRN, and find it is worse than NLRN by about 0.2 dB, showing the advantage of using non-local image properties for image restoration. Besides, we implement NLRNs where non-local module is used in every other state or every three states, and observe that if the frequency of using non-local modules in NLRN is reduced, the performance decreases accordingly. We show the benefit of propagating correlation information among adjacent states by comparing with the counter-part in terms of restoration accuracy. To further analyze the non-local module, we visualize the feature correlation maps for non-local operations in Figure 4. It can be seen that as the number of recurrent states increases, the locations

**Table 1:** Image denoising comparison of our proposed model with various distance metrics on Set12 with noise level of 25.

| Distance metric | $\phi(\boldsymbol{X}_i, \boldsymbol{X}_j)$ | PSNR |
|---|---|---|
| Euclidean distance | $\exp\{-\|\boldsymbol{X}_i - \boldsymbol{X}_j\|_2^2 / h^2\}$ | 30.74 |
| Dot product | $\boldsymbol{X}_i \boldsymbol{X}_j^T$ | 30.68 |
| Embedded dot product | $\theta(\boldsymbol{X}_i)\psi(\boldsymbol{X}_j)^T$ | 30.75 |
| Gaussian | $\exp\{\boldsymbol{X}_i \boldsymbol{X}_j^T\}$ | 30.69 |
| Symmetric embedded Gaussian | $\exp\{\theta(\boldsymbol{X}_i)\theta(\boldsymbol{X}_j)^T\}$ | 30.76 |
| Embedded Gaussian | $\exp\{\theta(\boldsymbol{X}_i)\psi(\boldsymbol{X}_j)^T\}$ | 30.80 |

**Table 2:** Image denoising comparison of our NLRN with its variants on Set12 with noise level of 25.

| Model | PSNR |
|---|---|
| NLRN w/o parameter sharing | 30.65 |
| RNN with same parameter no. | 30.61 |
| Non-local module in every other state | 30.76 |
| Non-local module in every 3 states | 30.72 |
| NLRN w/o propagating correlations | 30.78 |
| NLRN | 30.80 |

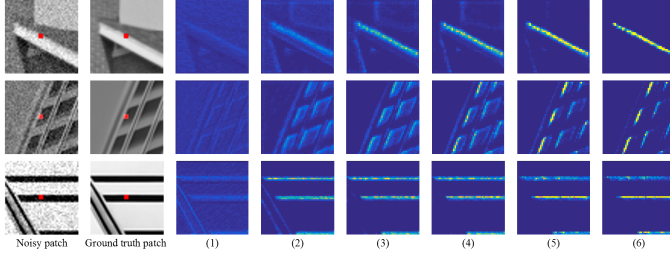
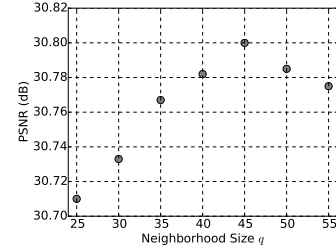

Noisy patch   Ground truth patch   (1)   (2)   (3)   (4)   (5)   (6)

**Figure 4:** Examples of correlation maps of non-local operations for image denoising. Noisy patch/ground truth patch: the neighborhood of the red center pixel used in non-local operations. (1)-(6): the correlation map for recurrent state 1-6 from NLRN with unrolling length of 6.

**Figure 5:** Neighborhood size vs. image denoising performance of our proposed model on Set12 with noise level of 25.

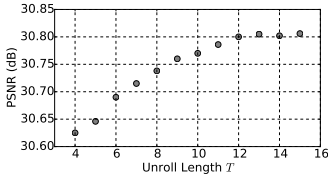

**Figure 6:** Unrolling length vs. image denoising performance of our proposed model on Set12 with noise level of 25.

|  | DnCNN | RED | MemNet | NLRN | | |
|---|---|---|---|---|---|---|
| Max effective depth | 17 | 30 | 80 | 38 | | |
| Parameter sharing | No | No | Yes | Yes | | |
| Parameter no. | 554k | 4,131k | 667k | 330k | | |
| Multi-view testing | No | Yes | No | No | No | Yes |
| Training images | 400 | 300 | 300 | 400 | 300 | 300 |
| PSNR | 27.18 | 27.33 | 27.38 | 27.64 | 27.60 | 27.66 |

**Table 3:** Image denoising comparison of our proposed model with state-of-the-art network models on Set12 with noise level of 50. Model complexities are also compared.

with similar features progressively show higher correlations in the map, which demonstrates the effectiveness of the non-local module for exploiting image self-similarity.

Figure 5 investigates the influence of the neighborhood size in the non-local module on image denoising results. The performance peaks at $q = 45$. This shows that limiting the neighborhood helps concentrate the correlation calculation on relevant features in the spatial vicinity and enhance correlation estimation. Therefore, it is necessary to choose a proper neighborhood size (rather than the whole image) for image restoration. We select $q = 45$ for the rest of this paper unless stated otherwise.

The unrolling length $T$ determines the maximum effective depth (*i.e.*, maximum number of convolutional layers) of NLRN. The influence of the unrolling length on image denoising results is shown in Figure 6. The performance increases as the unrolling length rises, but gets saturated after $T = 12$. Given the tradeoff between restoration accuracy and inference time, we adopt $T = 12$ for NLRN in all the experiments.

## 5.2   Comparisons with State-of-the-Art Methods

We compare our proposed model with a number of recent competitors for image denoising and image SR, respectively. PSNR and SSIM [42] are adopted for measuring quantitative restoration performance.

**Image Denoising**: For a fair comparison with other methods based on deep networks, we train our model under two settings: (1) We use the training data as in TNRD [7], DnCNN [49] and NLNet [23], and the result is shown in Table 4. We cite the result of NLNet in the original paper [23], since no public code or model is available. (2) We use the training data as in RED [28] and MemNet [36], and the result is shown in Table 5. We note that RED uses multi-view testing [43] to boost the restoration accuracy, *i.e.*, RED processes each test image as well as its rotated and flipped versions, and all the outputs are then averaged to form the final denoised image. Accordingly, we perform the same procedure for NLRN and find its performance, termed as *NLRN-MV*, is consistently improved. In addition, we include recent non-deep-learning based methods: BM3D [8] and WNNM [15] in our comparison. We do not list other methods [52, 3, 45, 6, 50] whose average performances are worse than DnCNN or MemNet. Our NLRN significantly outperforms all the competitors on Urban100 and yields the best results across almost all the noise levels and datasets.

To further show the advantage of the network design of NLRN, we compare different versions of NLRN with several state-of-the-art network models, *i.e.*, DnCNN, RED and MemNet in Table 3. NLRN uses the fewest parameters but outperforms all the competitors. Specifically, NLRN benefits

**Table 4:** Benchmark image denoising results. Training and testing protocols are followed as in [49]. Average PSNR/SSIM for various noise levels on Set12, BSD68 and Urban100. The best performance is in bold.

| Dataset | Noise | BM3D | WNNM | TNRD | NLNet | DnCNN | NLRN |
|---|---|---|---|---|---|---|---|
| Set12 | 15 | 32.37/0.8952 | 32.70/0.8982 | 32.50/0.8958 | -/- | 32.86/0.9031 | **33.16/0.9070** |
| | 25 | 29.97/0.8504 | 30.28/0.8557 | 30.06/0.8512 | -/- | 30.44/0.8622 | **30.80/0.8689** |
| | 50 | 26.72/0.7676 | 27.05/0.7775 | 26.81/0.7680 | -/- | 27.18/0.7829 | **27.64/0.7980** |
| BSD68 | 15 | 31.07/0.8717 | 31.37/0.8766 | 31.42/0.8769 | 31.52/- | 31.73/0.8907 | **31.88/0.8932** |
| | 25 | 28.57/0.8013 | 28.83/0.8087 | 28.92/0.8093 | 29.03/- | 29.23/0.8278 | **29.41/0.8331** |
| | 50 | 25.62/0.6864 | 25.87/0.6982 | 25.97/0.6994 | 26.07/- | 26.23/0.7189 | **26.47/0.7298** |
| Urban100 | 15 | 32.35/0.9220 | 32.97/0.9271 | 31.86/0.9031 | -/- | 32.68/0.9255 | **33.45/0.9354** |
| | 25 | 29.70/0.8777 | 30.39/0.8885 | 29.25/0.8473 | -/- | 29.97/0.8797 | **30.94/0.9018** |
| | 50 | 25.95/0.7791 | 26.83/0.8047 | 25.88/0.7563 | -/- | 26.28/0.7874 | **27.49/0.8279** |

**Table 5:** Benchmark image denoising results. Training and testing protocols are followed as in [36]. Average PSNR/SSIM for various noise levels on 14 images, BSD200 and Urban100. Red is the best and blue is the second best performance.

| Dataset | Noise | BM3D | WNNM | RED | MemNet | NLRN | NLRN-MV |
|---|---|---|---|---|---|---|---|
| 14 images | 30 | 28.49/0.8204 | 28.74/0.8273 | 29.17/0.8423 | 29.22/0.8444 | 29.37/0.8460 | 29.41/0.8472 |
| | 50 | 26.08/0.7427 | 26.32/0.7517 | 26.81/0.7733 | 26.91/0.7775 | 27.00/0.7777 | 27.05/0.7791 |
| | 70 | 24.65/0.6882 | 24.80/0.6975 | 25.31/0.7206 | 25.43/0.7260 | 25.49/0.7255 | 25.54/0.7273 |
| BSD200 | 30 | 27.31/0.7755 | 27.48/0.7807 | 27.95/0.8056 | 28.04/0.8053 | 28.15/0.8423 | 28.20/0.8436 |
| | 50 | 25.06/0.6831 | 25.26/0.6928 | 25.75/0.7167 | 25.86/0.7202 | 25.93/0.7214 | 25.97/0.8429 |
| | 70 | 23.82/0.6240 | 23.95/0.6346 | 24.37/0.6551 | 24.53/0.6608 | 24.58/0.6614 | 24.62/0.6634 |
| Urban100 | 30 | 28.75/0.8567 | 29.47/0.8697 | 29.12/0.8674 | 29.10/0.8631 | 29.94/0.8830 | 29.99/0.8842 |
| | 50 | 25.95/0.7791 | 26.83/0.8047 | 26.44/0.7977 | 26.65/0.8030 | 27.38/0.8241 | 27.43/0.8256 |
| | 70 | 24.27/0.7163 | 25.11/0.7501 | 24.75/0.7415 | 25.01/0.7496 | 25.66/0.7707 | 25.71/0.7724 |

**Table 6:** Benchmark SISR results. Average PSNR/SSIM for scale factor $\times 2$, $\times 3$ and $\times 4$ on datasets Set5, Set14, BSD100 and Urban100. The best performance is in bold.

| Dataset | Scale | SRCNN | VDSR | DRCN | LapSRN | DRRN | MemNet | NLRN |
|---|---|---|---|---|---|---|---|---|
| Set5 | $\times 2$ | 36.66/0.9542 | 37.53/0.9587 | 37.63/0.9588 | 37.52/0.959 | 37.74/0.9591 | 37.78/0.9597 | **38.00/0.9603** |
| | $\times 3$ | 32.75/0.9090 | 33.66/0.9213 | 33.82/0.9226 | 33.82/0.923 | 34.03/0.9244 | 34.09/0.9248 | **34.27/0.9266** |
| | $\times 4$ | 30.48/0.8628 | 31.35/0.8838 | 31.53/0.8854 | 31.54/0.885 | 31.68/0.8888 | 31.74/0.8893 | **31.92/0.8916** |
| Set14 | $\times 2$ | 32.45/0.9067 | 33.03/0.9124 | 33.04/0.9118 | 33.08/0.913 | 33.23/0.9136 | 33.28/0.9142 | **33.46/0.9159** |
| | $\times 3$ | 29.30/0.8215 | 29.77/0.8314 | 29.76/0.8311 | 29.79/0.832 | 29.96/0.8349 | 30.00/0.8350 | **30.16/0.8374** |
| | $\times 4$ | 27.50/0.7513 | 28.01/0.7674 | 28.02/0.7670 | 28.19/0.772 | 28.21/0.7721 | 28.26/0.7723 | **28.36/0.7745** |
| BSD100 | $\times 2$ | 31.36/0.8879 | 31.90/0.8960 | 31.85/0.8942 | 31.80/0.895 | 32.05/0.8973 | 32.08/0.8978 | **32.19/0.8992** |
| | $\times 3$ | 28.41/0.7863 | 28.82/0.7976 | 28.80/0.7963 | 28.82/0.797 | 28.95/0.8004 | 28.96/0.8001 | **29.06/0.8026** |
| | $\times 4$ | 26.90/0.7101 | 27.29/0.7251 | 27.23/0.7233 | 27.32/0.728 | 27.38/0.7284 | 27.40/0.7281 | **27.48/0.7306** |
| Urban100 | $\times 2$ | 29.50/0.8946 | 30.76/0.9140 | 30.75/0.9133 | 30.41/0.910 | 31.23/0.9188 | 31.31/0.9195 | **31.81/0.9249** |
| | $\times 3$ | 26.24/0.7989 | 27.14/0.8279 | 27.15/0.8276 | 27.07/0.827 | 27.53/0.8378 | 27.56/0.8376 | **27.93/0.8453** |
| | $\times 4$ | 24.52/0.7221 | 25.18/0.7524 | 25.14/0.7510 | 25.21/0.756 | 25.44/0.7638 | 25.50/0.7630 | **25.79/0.7729** |

from inherent parameter sharing and uses only less than 1/10 parameters of RED. Compared with the RNN competitor, MemNet, NLRN uses only half of parameters and much shallower depth to obtain better performance, which shows the superiority of our non-local recurrent architecture.

**Image Super-Resolution**: We compare our model with several recent SISR approaches, including SRCNN [10], VDSR [20], DRCN [21], LapSRN [22], DRRN [35] and MemNet [36] in Table 6. We crop pixels near image borders before calculating PSNR and SSIM as in [10, 33, 20, 21]. We do not list other methods [19, 33, 25, 34, 16] since their performances are worse than that of DRRN or MemNet. Besides, we do not include SRDenseNet [39] and EDSR [24] in the comparison because the number of parameters in these two network models is over two orders of magnitude larger than that of our NLRN and their training datasets are significantly larger than ours. It can be seen that NLRN yields the best result across all the upscaling factors and datasets. Visual results are provided in the supplementary material.

## 6 Conclusion

We have presented a new and effective recurrent network that incorporates non-local operations for image restoration. The proposed non-local module can be trained end-to-end with the recurrent network. We have studied the importance of computing reliable feature correlations within a confined neighborhood against the whole image, and have shown the benefits of passing feature correlation messages between adjacent recurrent stages. Comprehensive evaluations over benchmarks for image denoising and super-resolution demonstrate the superiority of NLRN over existing methods.

## Footnotes

[1] In our analysis, if $\boldsymbol{A}$ is a matrix, $\boldsymbol{A}_i$, $\boldsymbol{A}^j$, and $\boldsymbol{A}_i^j$ denote its $i$-th row, $j$-th column, and the element at the $i$-th row and $j$-th column, respectively.

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
