[Supplementary Material]

# Non-Local Recurrent Network for Image Restoration
# Supplementary Material

**Ding Liu**[1], **Bihan Wen**[1], **Yuchen Fan**[1], **Chen Change Loy**[2], **Thomas S. Huang**[1]

[1]University of Illinois at Urbana-Champaign    [2]Nanyang Technological University

{dingliu2, bwen3, yuchenf4, t-huang1}@illinois.edu  ccloy@ntu.edu.sg

## 1   Overview

In this supplementary document, we present additional results to complement the paper. First, we present an extension of our general framework to other classic non-local methods for image restoration. Second, we provide visual results for the comparison of our NLRN and several competing methods on image denoising and image super-resolution.

## 2   Extension of the General Framework to Other Classic Non-Local Methods

Besides the extension to WNNM and non-local means, which are discussed in Section 3.2 of the main paper, we show the proposed non-local framework can be extended to collaborative filtering methods, *e.g.*, BM3D algorithm [1], as well as joint sparsity based methods, *e.g.*, LSSC algorithm [6]. We follow the same notations in Section 3.2 of the main paper. Both BM3D and LSSC apply block matching (BM) first before processing, and form $N$ groups of similar patches into data matrices. The index set of the matched patches for the $i$-th reference patch is denoted as $\mathbb{C}_i$. The group of matched patches for the $i$-th reference patch is denoted as $\boldsymbol{X}_{\mathbb{C}_i}$.

Similar to WNNM [3], BM3D [1] also applies BM first to group similar patches based on their Euclidean distances. The matched patches are then processed via Wiener filtering [1], and the denoised results of the $i$-th group of patches are

$$\boldsymbol{Z}_{\mathbb{C}_i} = \tau^{-1}(\text{diag}(\omega)\tau(\boldsymbol{X}_{\mathbb{C}_i})). \qquad (1)$$

Here $\tau(\cdot)$ and $\tau^{-1}(\cdot)$ denote the forward and backward Wiener filtering applied to the groups of matched patches, respectively. The diagonal matrix $\text{diag}(\omega)$ is formed by the empirical Wiener coefficients $\omega$. BM3D applies data pre-cleaning, using discrete cosine transform (DCT), to estimate the original patch, and calculate the estimate of $\omega$ [1]. Since calculating $\boldsymbol{Z}_{\mathbb{C}_i}$ in (1) involves only linear filtering, it can also be generalized using the proposed non-local framework. Unlike the extension to WNNM, here $\sum_{j \in \mathbb{C}_i} \Phi(\boldsymbol{X})_i^j \boldsymbol{G}(\boldsymbol{X})_j$ corresponds to the denoised results via Wiener filtering as shown in (1), of the $i$-th group of matched patches.

Different from BM3D and WNNM, LSSC learns a common dictionary $\boldsymbol{D}$ for all image patches, and imposes joint sparsity [6] on each data matrix of matched patches $\boldsymbol{X}_{\mathbb{C}_i}$, so that the correlation of the matched patches are exploited by enforcing the same support of their sparse codes. Thus, the joint sparse coding in LSSC [6] becomes

$$\hat{\boldsymbol{A}}_i = \text{argmin}_{\boldsymbol{A}_i} \|\boldsymbol{A}_i\|_{0,\infty} \quad s.t. \quad \left\| \boldsymbol{X}_{\mathbb{C}_i}^T - \boldsymbol{D}\boldsymbol{A}_i \right\|_F^2 \leq \epsilon |\mathbb{C}_i|, \ \forall i \,, \qquad (2)$$

where the $(0,\infty)$ "norm" $\|\cdot\|_{0,\infty}$ counts the number of non-zero columns of each sparse code matrix $\boldsymbol{A}_i$ [6], and $|\mathbb{C}_i|$ is the cardinality of $\mathbb{C}_i$. The coefficient $\epsilon$ is a constant, which is used to upper bound the sparse modeling errors. In general, the solution to (2) is NP-hard. To simplify the discussion, we assume the dictionary to be unitary (which reduces the sparse coding problem to the transform-model

sparse coding [10]), *i.e.*, $\boldsymbol{D}^T \boldsymbol{D} = \boldsymbol{I}$ and $\boldsymbol{D} \in \mathbb{R}^{k \times k}$. Thus there exists a corresponding shrinkage function $\eta(\cdot)$ for imposing joint sparsity on the sparse codes [6, 7], such that the denoised estimates of the $i$-th patch group can be obtained as $\boldsymbol{Z}_{\mathbb{C}_i} = \hat{\boldsymbol{A}}_i^T \boldsymbol{D}^T = \eta(\boldsymbol{X}_{\mathbb{C}_i} \boldsymbol{D}) \boldsymbol{D}^T$. Though joint sparse coding projects all data onto a union of subspaces [6, 2, 10] which is a non-linear operation in general, each data matrix $\boldsymbol{X}_{\mathbb{C}_i}$ is projected onto one particular subspace spanned by the selected atoms corresponding to the non-zero columns in $\hat{\boldsymbol{A}}_i$, which is locally linear. For the $i$-th group of patches, such a subspace projection corresponds to $\sum_{j \in \mathbb{C}_i} \Phi(\boldsymbol{X})_i^j \boldsymbol{G}(\boldsymbol{X})_j$ in the proposed general framework.

## 3  Visual Results

We show the visual comparison of our NLRN and several competing methods: BM3D [1], WNNM [3], and MemNet [9] for image denoising in Figure 1. Our method can recover more details from the noisy measurement. The visual comparison of our NLRN and several recent methods: DRCN [4], LapSRN [5], DRRN [8], and MemNet [9] for image super-resolution is displayed in Figure 2. Our method is able to reconstruct sharper edges and produce fewer artifacts especially in the regions of repetitive patterns.

**Figure 1:** Qualitative comparison of image denoising results with noise level of 30. The zoom-in region in the red bounding box is shown on the right. From top to bottom: 1) the image *barbara*. 2) image *004* in Urban100. 3) image *019* in Urban100. 4) image *033* in Urban100. 5) image *046* in Urban100.

**Figure 2:** Qualitative comparison of image super-resolution results with ×4 upscaling. The zoom-in region in the red bounding box is shown on the right. From top to bottom: 1) image *005* in Urban100. 2) image *019* in Urban100. 3) image *044* in Urban100. 4) image *062* in Urban100. 5) image *099* in Urban100.