[Reviews · NeurIPS 2018]

Reviewer 1



This paper introduced a mathematical framework to describe nonlocal operations in image restoration, such as nonlocal means, low-rank models, etc. Then the authors use the framework as a module of an RNN and validated the network on a few datasets for image denoising and super-resolution. Overall, I see merits of this paper, while I have a few questions/concerns. (See below.) Pros: 1) The proposed framework is a nice summary of exsiting nonlocal operations commonly used in image restoration. 2) The proposed deep model works epseically well on texture images, and has advantage over several exsiting models. Cons: 1) It has already known to the literature that some nonlocal operations can be put in matrix/operator forms. The authors claims they were the first to propose such framework. However, I think they should address the relation and differences between their framework and existing ones, such as the ones listed below: -- Yin, Rujie, et al. "A tale of two bases: Local-nonlocal regularization on image patches with convolution framelets." SIAM Journal on Imaging Sciences 10.2 (2017): 711-750. -- Danielyan, Aram, Vladimir Katkovnik, and Karen Egiazarian. "BM3D frames and variational image deblurring." IEEE Transactions on Image Processing 21.4 (2012): 1715-1728. 2) The authors emphasized a multiple places as one of their contributions that local search regions for similarity should be used. This is one of the main differences between their model with [37]. However, such treatment is a common practice for nonlocal means. The other two differences from [37] seems minor. From the title of the paper, the non-localness is the main selling point of the paper, which has already appeared in [37]. 3) In the mathematical framework, the use of correlation features is not included. However, it appears in RNN model. From Table 2, with or without correlation propergation does not seem to make much difference. Then why bother introducing the additional path of s_corr in the network? I might have misinterpreted the paper, but in any case, I think this should be further clarified. 4) How does the proposed network compare with some exsiting nonlocal+dl denoising models such as Lefkimmiatis, Stamatios. "Non-local color image denoising with convolutional neural networks." In Proc. IEEE Int. Conf. Computer Vision and Pattern Recognition, pp. 3587-3596. 2017.

Reviewer 2



Conceptually, it makes sense to use an RNN to denoise images. Using non-local information is conjectured to help in denoising, as shown by the experiments. There is however a bit confusion in this work between image restoration and image denoising/super-resolution. Image restoration is defined as the task which consists of inverting a point spread function. This operation is useful in a wide array of domains ranging from astronomy to microscopy. Image denoising can be viewed as the simple case when this PSF is equal to identity and image super-resolution corresponds to a decimated version of this identity operator (so that it does no longer consist of a convolutive model). There is no evidence that the approach presented in this work applies to more general problems than denoising and super-resolution. Qualitative results are missing in the paper, so we do not have any idea of how the results are improved visually. At the same time, the overall improvements are not ground-breaking in terms of PSNR values

Reviewer 3



The authors build on top of the non-local neural network of Wang et al., 2017 [37] and incorporate it into a recurrent neural network for image restoration. The non-local recurrent network (NLRN) provides top results on image denoising and super-resolution. The paper reads well and provides a good balance between method description and experimental results. The reported results are very good especially for the image denoising task. For the image super-resolution study I suggest the following works: Timofte et al., NTIRE 2017 challenge on single image super-resolution: Methods and results, 2017 Ledig et al., Photo-realistic single image super-resolution using a generative adversarial network, 2017 Bae et al., Beyond deep residual learning for image restoration: Persistent homology-guided manifold simplification, 2017 I think that the authors should compare with the approaches of Bae et al., 2017 and of Ledig et al., 2017 (SRResNet). A runtime / time complexity comparison, in addition to the Table 3 contents, would be interesting to have.